# To Hemoadsorb or Not to Hemoadsorb—Do We Have the Answer Yet? An Updated Meta-Analysis on the Use of CytoSorb in Sepsis and Septic Shock

**DOI:** 10.3390/biomedicines13010180

**Published:** 2025-01-13

**Authors:** Carmen Orban, Angelica Bratu, Mihaela Agapie, Tudor Borjog, Mugurel Jafal, Romina-Marina Sima, Oana Clementina Dumitrașcu, Mihai Popescu

**Affiliations:** 1Obstetrics and Gynecology, Anesthesia and Intensive Care, Department 14, School of Medicine, “Carol Davila” University of Medicine and Pharmacy, 37 Dionisie Lupu Street, 020021 Bucharest, Romania; carmen.orban@umfcd.ro (C.O.); agapiemili@yahoo.com (M.A.); tudorborjog@gmail.com (T.B.); mugureljafal@yahoo.com (M.J.); romina.sima@umfcd.ro (R.-M.S.); mihai.popescu@umfcd.ro (M.P.); 2Bucharest University Emergency Hospital, 169 Splaiul Independentei, 050098 Bucharest, Romania; oanadumit@gmail.com; 3“Bucur” Maternity, “Saint John” Hospital, 040294 Bucharest, Romania

**Keywords:** sepsis, septic shock, hemoadsorption, CytoSorb, renal replacement therapy

## Abstract

Severe inflammation leading to organ dysfunction is the cornerstone of the pathophysiology of sepsis. Thus, from a theoretical point of view, rebalancing inflammation has the potential to improve patient outcomes. Methods: To better understand the clinical effectiveness of hemoadsorption in managing inflammation, we conducted an updated meta-analysis on the effects of CytoSorb in critically ill septic patients. Ten studies containing 715 patients (355 in the interventional group and 360 in the control group) have been included in the final analysis. Results: Statistical analysis demonstrated that the use of CytoSorb did not influence overall mortality (OR 0.95, 95% CI [0.58, 1.56], *p* = 0.85), but we observed a decreased mortality when comparing CytoSorb-treated patients with patients in the control group treated with continuous renal replacement therapy (CRRT) (OR 0.97, 95% CI [0.46, 0.98], *p* = 0.04). We also observed an increased mortality in patients in whom hemoadsorption was initiated earlier in the treatment course (OR 0.97, 95% CI [0.46, 0.98], *p* = 0.04). We did not observe any significant difference in either intensive care unit length of stay (*p* = 0.93) or between end-of-treatment severity scores in the two groups (*p* = 0.24). Conclusions: Although it has a high risk of bias, current evidence does not support the routine use of CytoSorb in critically ill septic patients. The addition of CytoSorb to CRRT may be associated with decreased survival as compared to CRRT alone, but future studies are needed to draw a definitive conclusion.

## 1. Introduction

The definition and management of sepsis and septic shock have changed dramatically over the last decade. It is now generally accepted that sepsis is more than a systemic inflammatory response syndrome secondary to an infection but rather a dysregulated immune response that results in life-threatening organ dysfunction(s) [1,2]. Despite all these improvements, mortality remains high in septic patients and has a wide geographic variation based on different socio-economic factors [3,4]. Current management of sepsis is centered on fluid management, early antimicrobial therapy, source control, and advanced organ support [5,6,7]. However, we should not forget that sepsis is an inflammatory disease, and rebalancing inflammation may be the skeleton key we are looking for in improving patient outcomes [8]. In fact, managing inflammation in septic patients has been one of the main topics of research in recent years [9,10,11]. However, due to conflicting results, current guidelines failed to offer a high-grade recommendation for immunomodulation.

CytoSorb was introduced in clinical practice as an extracorporeal organ support device more than ten years ago with the theoretical purpose of eliminating hydrophobic molecules with a molecular mass of up to 55 kDa [12]. Its potential benefits have been proposed by observational studies in sepsis, acute respiratory distress syndrome, liver failure, and any disease that is associated with a hyperinflammatory state [13,14,15]. Although CytoSorb is generally considered a safe procedure with minimal additional risks, there is no consensus on its efficacy [16]. However, with the growing evidence and increasing quality of CytoSorb research studies, our updated meta-analysis tries to provide a better understanding of the clinical use of this hemoadsorption device in critically ill septic patients.

## 2. Materials and Methods

This study aimed to perform an updated meta-analysis to assess the potential benefits of hemoadsorption with CytoSorb in the management of patients with sepsis or septic shock. For the uniformity of included studies, sepsis/septic shock and the standard of care were those defined by recently published guidelines [1,2].

The primary outcome of our meta-analysis was to assess the effects of hemoadsorption on mortality in septic patients. The secondary outcomes were to determine the effects of hemoadsorption on (1) the dynamics of the SOFA score and (2) hospital length of stay.

To assess our objectives, we included randomized and non-randomized controlled trials in which hemoadsorption by CytoSorb was used either alone or in combination with any renal replacement therapy (RRT) and was compared with either standard of care or other RRTs. Studies published as abstracts only were excluded. However, if an international conference abstract was identified as matching the inclusion criteria for our meta-analysis, we contacted the corresponding author for further information on the publication status of their research as an original article. To address a broader picture of sepsis that can be found in any intensive care unit (ICU) setting, we excluded studies in which CytoSorb was applied on an extracorporeal membrane oxygenation (ECMO) circuit or in patients who were already on ECMO. Also, we excluded studies in which infectious endocarditis was the cause of sepsis or in which patients had undergone cardiac surgery. In case the inclusion or exclusion criteria of selected studies, as well as the therapy applied, were unclear, we contacted the corresponding author to clarify the ambiguity.

The following databases were searched by two reviewers (C.O. and A.B.): PubMed, EMBASE, Scopus, and Web of Science. The timeline was set from 2016 to September 2024 to comply with the current definition of sepsis. Language restrictions were set to English only. The following terms were used in the search strategy: CytoSorb, hemadsorption, sepsis, septic shock, mortality, length of hospital stay, and SOFA score. We also searched the reference list of previously published meta-analyses and systematic reviews to identify published articles not indexed in the databases we searched.

The inclusion and exclusion criteria were applied to the abstracts identified in the database search and all duplicates were removed. After this initial step, the full-length version of the included trials was screened by the two reviewers (C.O. and A.B.). In cases of disagreement between the two reviewers, a third reviewer (M.P.) was included in the decision making. The present meta-analysis was performed in accordance with the Reporting Items for Systematic Reviews and Meta-Analysis Guidelines (PRISMA) statement guidelines for conducting and reporting meta-analyses and the PICOS scheme for reporting inclusion criteria. The risk of bias was independently performed by two reviewers (M.A. and T.B.) using the tool provided by the Cochrane Collaboration (London, United Kingdom) [17].

After the final list of included articles was approved by all the authors, data were extracted in a standardized form by two reviewers (M.J. and R.M.S.) and compared for discrepancies at the end of the process. We collected the following information: basic information (author name, year of publication, type of hospital, country in which the study was performed), number of patients, age, SOFA and/or APACHE II score, treatment therapy, duration of therapy, number of CytoSorb sessions, time of initiation of the therapy, primary outcome (patients’ survival), and secondary outcomes (final SOFA score, length of hospital stay).

Statistical analysis. The statistical analysis was performed using the Rev-Man online tool (Cochrane Collaboration, Oxford, UK) available at https://revman.cochrane.org/ (accessed 20 September 2024). Results are expressed as odds ratio (OR) for dichotomous data or weighted mean differences for continuous data, both with 95% confidence intervals (CI). Statistical heterogeneity was quantified using the I^2^ statistic. An I^2^ > 50% was considered to indicate substantial heterogeneity. In this case, we conducted a random-effect meta-analysis; otherwise, we used the fixed-effect model in cases of non-heterogeneity. For effect sizes, the odds ratio (OR) for dichotomous outcomes and standardized mean difference (SMD) for continuous variables were calculated using a random-effect model in cases of significant heterogeneity between estimates. A *p*-value > 0.05 was considered to reject the null hypothesis that the studies were heterogeneous.

## 3. Results

Our initial database search identified 506 published trials and abstracts. After the initial screening, 86 articles were included in the full assessment step. Of these, 76 were excluded due to various reasons. We finally identified 10 studies [18,19,20,21,22,23,24,25,26,27] that fulfilled the inclusion criteria to be included in our meta-analysis. The steps of our study selection are presented in Figure 1.

Of the ten included studies, three were randomized control studies (RCTs) and seven were case–control studies. The treatment arm in all studies consisted of continuous renal replacement therapy (CRRT) in combination with CytoSorb, while the control arm consisted of standard medical care in three studies and CRRT alone in seven studies. The commencement of therapy ranged widely between studies from early initiation of the time of diagnosis of septic shock [24] to up to 30 days [21]. The Cochrane risk-of-bias tool was used to assess the quality of the included RCTs and their associated risk of bias (Figure 2).

A total of 715 patients were included in the meta-analysis, with 355 in the interventional group and 360 in the control group. The mean age was around 60 years in all studies, and the mean SOFA score was between 12 and 14. A summary of included studies is presented in Table 1.

### 3.1. Mortality Assessment

Regarding overall mortality assessment, the I2 was 57%, demonstrating significant heterogeneity, and we performed a random-effects analysis. No significant difference was observed between the CytoSorb group (53.2%) compared to the control group (53.8%) in terms of mortality (OR 0.95, 95% CI [0.58, 1.56], *p* = 0.85)—Figure 3.

We further tested the effects of CytoSorb compared to standard CRRT. A total of six studies (five case–control and one randomized control trial) were included in this sub-analysis, with 230 patients in the hemoadsorption group and 232 patients in the control group. Statistical analysis demonstrated decreased mortality in CytoSorb-treated (50.8%) patients compared to CRRT-alone (60.7%) patients (OR 0.97, 95% CI [0.46, 0.98], *p* = 0.04)—Figure 4.

Three studies were identified as fulfilling the criteria of the early initiation of CytoSorb and were included in the subgroup analysis. Of these, two were randomized control trials, and one was a case–control matched analysis. Statistical analysis demonstrated increased mortality in the early initiation of CytoSorb treatment (67.9%) compared to the controls (52.3%) (OR 0.97, 95% CI [0.46, 0.98], *p* = 0.04)—Figure 5.

### 3.2. ICU Length of Stay and Severity Scores

Three studies assessed ICU length of stay and included 129 patients in the CytoSorb group and 133 patients in the control group. As the I2 was 93%, it demonstrated significant heterogeneity, and we performed a random-effects analysis. There was no difference between the CytoSorb group and the control group in terms of ICU length of stay (OR −0.38, 95% CI [−8.57, 7.82], *p* = 0.93)—Figure 6.

Four studies reported on the effects of CytoSorb on the SOFA score. Statistical analysis demonstrated that there was a significant difference in the mean SOFA score in favor of the control group compared to the SOFA group at the time of therapy initiation (OR 0.70, 95% CI [0.28, 1.12], *p* < 0.01)—Figure 7A. However, we did not find a statistically significant difference in the SOFA score between the two groups at the time of ICU discharge (OR 0.47, 95% CI [−0.32, 1.25], *p* = 0.24)—Figure 7B.

## 4. Discussion

CytoSorb has gained increased popularity as an extracorporeal support device that can rebalance the inflammatory response in sepsis. Although its potential benefits have been argued by different observational studies [28], some of the recently published randomized control trials failed to reach this conclusion [29]. This might be due to the vast heterogeneity in study designs ranging from the underlying disease, time of initiation, intensity of treatment, and duration of therapy. To address some of these issues, we tried to perform a meta-analysis of published evidence in septic patients alone, thus focusing on a single disease but with a complex and variable pathophysiology. In our methodology, we decided not to include studies in which infectious endocarditis requiring heart surgery was the source of sepsis or those studies in which patients were on extracorporeal membrane oxygenation due to the existing evidence supporting the effects of cardiopulmonary bypass on the inflammatory response [30,31]. In doing so, we tried to address a more “standard critically ill septic patient” that can be found in any general ICU.

Our results failed to demonstrate a beneficial effect of CytoSorb therapy in septic patients. The same conclusion was reached by previous, more heterogenic, meta-analysis. Becker et al. [14] included 34 studies published between 2010 and 2022, with more than 1000 treated patients in each arm and failed and did not observe an improved survival either in the entire cohort or in subgroup analyses of sepsis, cardiopulmonary bypass surgery, or critical illness. Moreover, Heymann et al. [29] reported higher mortality at the latest follow-up and at 30 days.

So, one of the key questions is why we have these conflicting results compared to clinical observations [32], and the answers are not easy. On the one hand, we have the significant heterogeneity of patient populations we have already mentioned. This heterogeneity does not refer just to the treatment itself (when and how CytoSorb was initiated), nor to the heterogeneity of included patients (different diseases that have their pathophysiology centered around a hyperinflammatory syndrome) but also to the varying degrees of inflammation septic patients may have during the course of illness [33]. As pro-inflammatory and anti-inflammatory cytokines are seldom measured in most of the published research before CytoSorb initiation, we cannot estimate the precise severity of the inflammatory syndrome. On the other hand, we know that this dysregulated host response may be responsible for organ dysfunction and mortality in sepsis [34] and that CytoSorb effectively diminishes circulating cytokine levels during systemic inflammation in humans [35]. What we fail to know is to what extent this “rebalanced” immune function has a positive effect on the body’s defense system against the invading micro-organism. The truth is that the precise interplay between pro- and anti-inflammatory cytokines in sepsis still eludes us.

The third problem is that CytoSorb exhibits a concentration-dependent removal of various substances [36]. It can be loosely translated that the adsorption effects of CytoSorb depend not only on the duration of treatment but also on the kinetics of cytokines (from initial concentrations to dynamics of cytokine production). So, a similar removal of different molecules in patients with varying concentrations cannot be assumed to be similar. Although our results did not support the benefits on overall mortality, we did observe an increased survival in patients treated with CytoSorb compared to those undergoing CRRT. The reason for this might be the more standardized approach to these patients in terms of the initiation of the therapy and CRRT dose. However, of the seven included studies, six were case–control with a high risk of bias, and only one was a randomized control trial. In that trial, Stockmann et al. [24] analyzed patients with Coronavirus disease (COVID-19) and failed to demonstrate either the resolution of vasoplegic shock or a decrease in mortality, inflammatory markers, and incidence of adverse events.

One important issue that needs to be addressed is the type of medical facility and the geographic area in which the included studies were conducted. These two factors may determine a significant variation in both the type of sepsis and the response to therapy. However, all studies, except one which was conducted in India, came from European countries. Also, except for a multicenter study from Germany in which the authors offered no information on the type of facility, all other medical units were either tertiary hospitals or university hospitals. Because of these, we considered that the quality and standards of medical care were similar between the studies and no subsequent analysis could be performed.

Antibiotic treatment represents one of the cornerstones of sepsis management. Both the appropriate antibiotic type [37] and obtaining adequate plasmatic concentration [38] are crucial for increased survival. Different studies have focused on the effects of CytoSorb on antibiotic pharmacokinetics. In one in vitro study [39], all tested drugs were adsorbed by CytoSorb in relevant amounts, especially during the first hours of therapy. The authors concluded that this may lead to the necessity of administering additional doses to counteract this effect, as well as drug concentration monitoring. This conclusion has also been observed in other clinical studies [40,41]. None of the included studies in our meta-analysis measured the plasmatic concentration of administered antibiotics, and this might be another reason for the non-significant difference in mortality, as the potential benefits of immunomodulation may be superseded by underoptimal antibiotherapy.

Our meta-analysis has some significant limitations that have been discussed above. First of all, CytoSorb is not a well-recognized and standardized therapy in patients with sepsis and septic shock. Although many studies have been published on this topic, most of them have a high risk of bias, the number of included patients is rather small, and there is no uniformity in either the indication, timing, or dosing of the therapy. Future research should focus on a more standardized approach to this therapy if definitive, high-quality evidence is sought out. These studies should focus on two important issues: (1) timing of therapy and (2) dosing of therapy. In terms of therapy initiation, an early approach (like within one hour from the diagnosis of sepsis, as in the case of antibiotics) would be better than the current literature, which left it at the discretion of the attending physician. Secondary, studies should focus on the duration of a single session, as well as the number of consecutive sessions applied. Currently, there is no consensus on this issue, and each center uses its own criteria for the duration and intensity of the therapy. A more objective approach, based on the dynamics of inflammatory markers, should be sought out.

## 5. Conclusions

Despite not being the “new kid on the block”, current evidence for the use of CytoSorb in patients with sepsis and septic shock does not support its routine use but rather a more personalized approach in patients who have severe systemic inflammation. Although it has a high risk of bias, the current evidence does not support the routine use of CytoSorb in critically ill septic patients. The addition of CytoSorb to CRRT may be associated with decreased survival as compared to CRRT alone, but future studies are needed to draw a definitive conclusion.

## Figures and Tables

**Figure 1 biomedicines-13-00180-f001:**
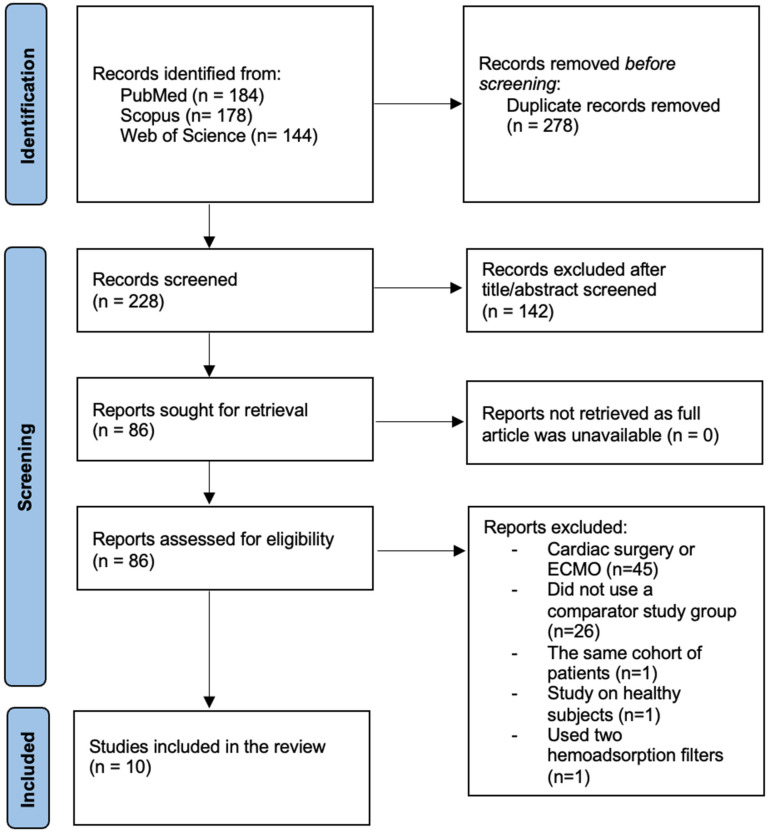
PRISMA flowchart of study selection based on the inclusion and exclusion criteria. Legend: ECMO—extracorporeal membrane oxygenation.

**Figure 2 biomedicines-13-00180-f002:**
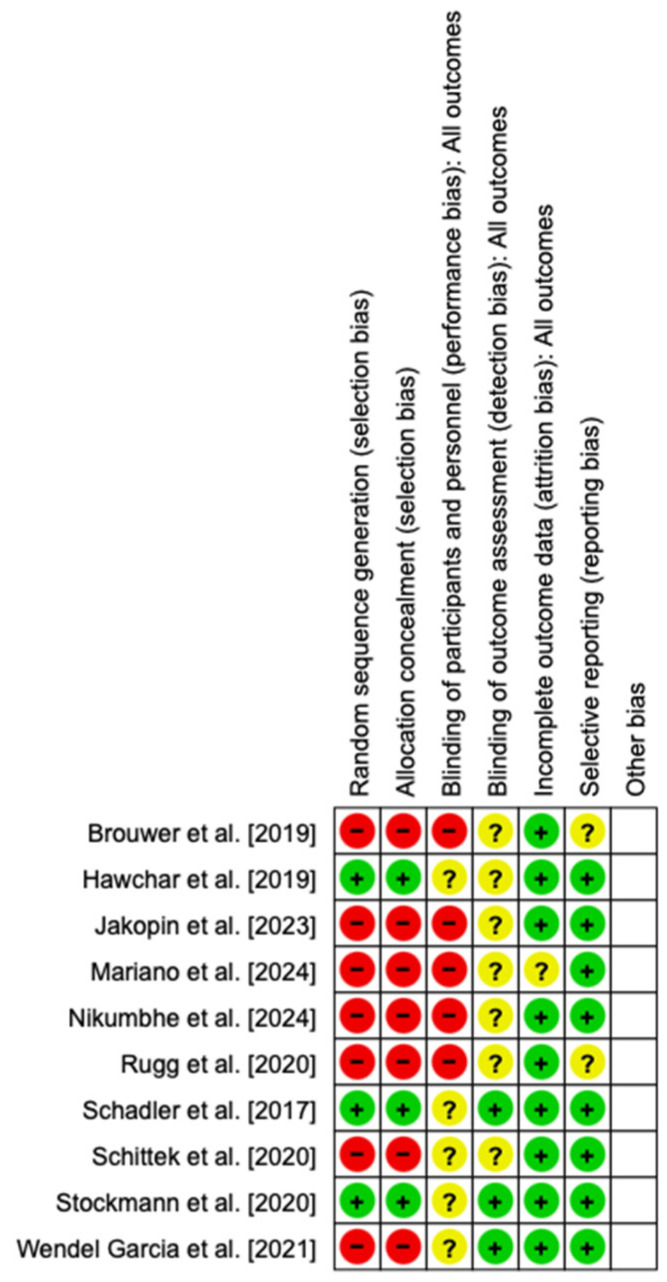
Summary of risk of bias. Brouwer et al. [2019] [20], Hawchar et al. [2019] [19], Jakopin et al. [2023] [25], Mariano et al. [2024] [26], Nikumbhe et al. [2024] [27], Rugg et al. [2020] [21], Schadler et al. [2017] [18], Schittek et al. [2020] [22], Stockmann et al. [2022] [24], Wendel Garcia et al. [2021] [23].

**Figure 3 biomedicines-13-00180-f003:**
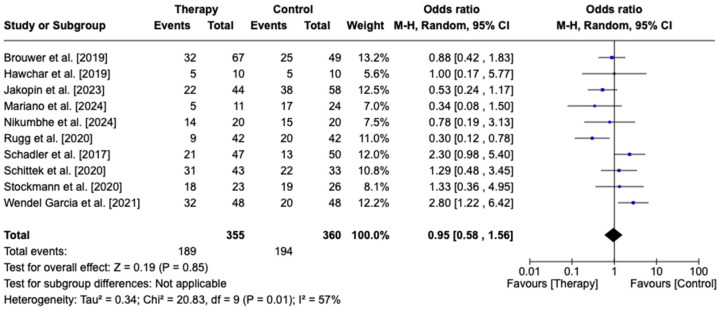
Forest plot for overall mortality comparing CytoSorb therapy with controls. Brouwer et al. [2019] [20], Hawchar et al. [2019] [19], Jakopin et al. [2023] [25], Mariano et al. [2024] [26], Nikumbhe et al. [2024] [27], Rugg et al. [2020] [21], Schadler et al. [2017] [18], Schittek et al. [2020] [22], Stockmann et al. [2022] [24], Wendel Garcia et al. [2021] [23].

**Figure 4 biomedicines-13-00180-f004:**
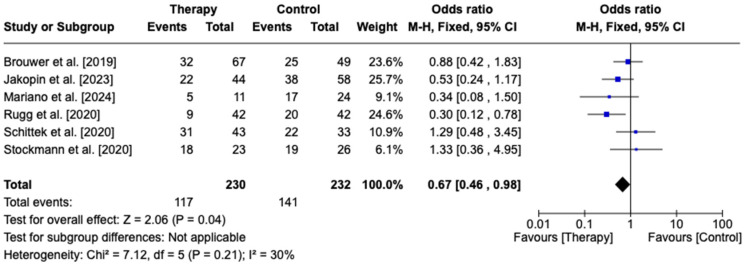
Forest plot comparing CytoSorb with continuous renal replacement therapy in septic patients. Brouwer et al. [2019] [20], Jakopin et al. [2023] [25], Mariano et al. [2024] [26], Rugg et al. [2020] [21], Schittek et al. [2020] [22], Stockmann et al. [2022] [24].

**Figure 5 biomedicines-13-00180-f005:**
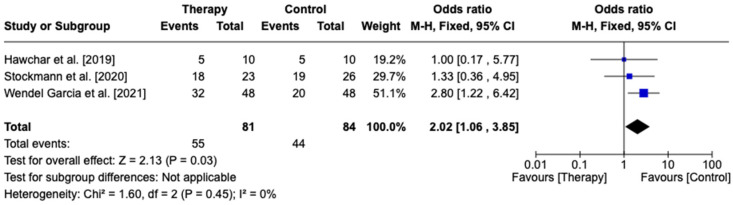
Forest plot comparing early initiation of CytoSorb-treated patients with controls. Hawchar et al. [2019] [19], Stockmann et al. [2022] [24], Wendel Garcia et al. [2021] [23].

**Figure 6 biomedicines-13-00180-f006:**
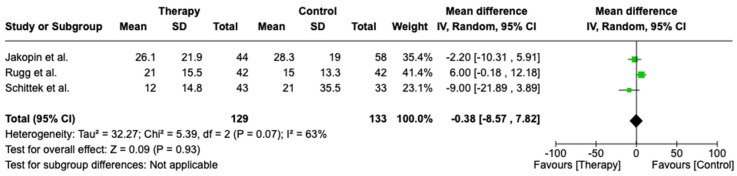
Forest plot comparing ICU length of stay between the CytoSorb group and the control group. Jakopin et al. [2023] [25], Rugg et al. [2020] [21], Schittek et al. [2020] [22].

**Figure 7 biomedicines-13-00180-f007:**
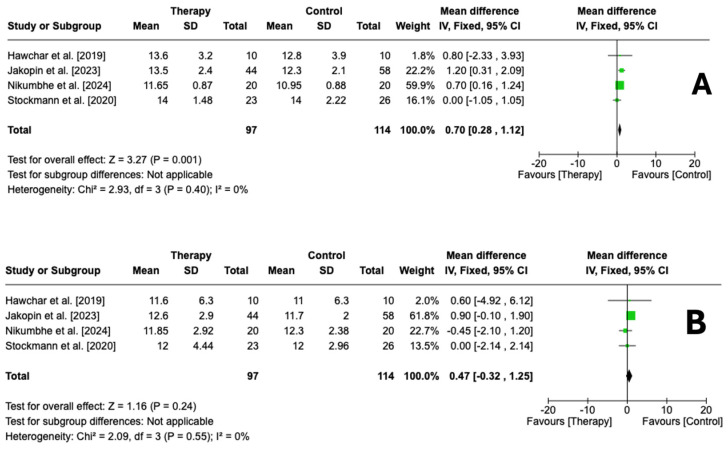
Forest plot comparing initial SOFA scores (**A**) and final SOFA scores (**B**) between the CytoSorb group and the control group. Hawchar et al. [2019] [19], Jakopin et al. [2023] [25], Nikumbhe et al. [2024] [27], Stockmann et al. [2022] [24].

**Table 1 biomedicines-13-00180-t001:** Characteristics of included studies and therapies applied.

Publication	Type of Study	Type of Hospital	Country of Origin	Patients: Total (CS/C)	Age (CS/C)	SOFA/APACHE II Score (CS/C)	Treatment Therapy	Control	Commencement of Therapy	Mortality Outcome (CS/C)
[18]	Open-label, RCT	NA	Germany	97 (47/50)	66/65	NR	CytoSorb hemoperfusion for 6 h/day for up to 7 days	Standard medical care	Within 72 h of diagnosis	44.7%/26.0%
Hawchar et al. [2019] [19]	RCT	UH	Hungary	20 (10/10)	60/71	13.6/12.8	24 h CRRT + CytoSorb	Standard medical care	Within 24 h of diagnosis	50.0%/50.0%
Brouwer et al. [2019] [20]	Case–control	UH	Nederland	116 (67/49)	61/68	11.7/11.8	CRRT + CytoSorb	CRRT	NR	47.8%/51.0%
Rugg et al. [2020] [21]	Case–control matched analysis	UH	Austria	84 (42/42)	64/68	13.0/12.0	CRRT + CytoSorb	CRRT	Up to 719 h (median 21.4 h) after ICU admission	21.4%/47.6%
Schittek et al. [2020] [22]	Case–control	TH	Austria	76 (43/33)	63/62	35.0/39.0	CVVHDF + CytoSorb	CVVHDF	NR	72.1%/66.7%
Wendel Garcia et al. [2021] [23]	Case–control matched analysis	UH	Switzerland	96 (48/48)	58/57	14.0/14.0	CytoSorb for 3 consecutive 24 h sessions	Standard medical care	Within 24 h of diagnosis	67.0%/42.0%
Stockmann et al. [2022] [24]	Open-label, RCT	UH	Germany	49 (23/26)	61/66	14.0/14.0	CRRT + CytoSorb for 3 to 7 24 h sessions	CRRT	After diagnosis of septic shock	78.0%/73.0%
Jakopin et al. [2023] [25]	Case–control	TH	Slovenia	102 (44/58)	64/70	NR	CVVH + CytoSorb	CVVH	NR	50.0%/65.5%
Mariano et al. [2024] [26]	Case–control	UT	Italy	35 (11/24)	63/72	12/12	CRRT + CytoSorb	CRRT	After 72 h of diagnosis of sepsis	45.4%/70.8%
Nikumbhe et al. [2024] [27]	Case–control	TH	India	40 (20/20)	45/51	10.9/ 11.6	CRRT/SLED + CytoSorb	Standard medical care + CRRT/SLED	Not reported	70.0%/75.0%

Legend: CS—CytoSorb; C—control; TH—tertiary hospital; UH—university hospital; SOFA—Sequential Organ Failure Assessment; APACHE II—acute physiology and chronic health evaluation II; RCT—randomized control trial; CRRT—continuous renal replacement therapy; CVVHDF—continuous veno-venous hemodiafiltration; CVVH—continuous veno-venous hemofiltration; NA—not available.

## Data Availability

Data are available by contacting the corresponding author.

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
