# Peer review of "To Hemoadsorb or Not to Hemoadsorb—Do We Have the Answer Yet? An Updated Meta-Analysis on the Use of CytoSorb in Sepsis and Septic Shock"

_biomedicines, 2025, doi:10.3390/biomedicines13010180_

Round 1
Reviewer 1 Report (Previous Reviewer 3)
Comments and Suggestions for Authors
The revised version has been considerably improved. Hence the same can be considered for publication in this journal of repute.
Author Response
Dear reviewer,
Thank you very much once again for your input you have previously provided to improve our previous version.
Reviewer 2 Report (Previous Reviewer 2)
Comments and Suggestions for Authors
Satisfactory revision
Author Response
Dear reviewer,
Thank you very much once again for the input you have provided for our previous version.
Reviewer 3 Report (Previous Reviewer 1)
Comments and Suggestions for Authors
Comments
I have reviewed the article title “To hemoadsorb or not to hemoadsorb – do we have the answer yet? An updated meta-analysis on the use of CytoSorb in sepsis and septic shock” and I have found some major flaws which need to be revised before further processing.
Abstract:
Please mention which tools/softwares were used for the data analysis and interpretation.
Line 25. Please revise as “We did not observe any significant difference…”
Line 62. Please edit the term Secondary” to “secondary”
Line 62. It would be much better if the timeline was set from 2014 to September 2024, or give a possible justification for time 2016-2024 selection.
Please providet the primary outcome datasheet of meta-analysis for assesing the effects of hemoadsorption on mortality in septic patients, the secondary outcomes were to determine the effects of hemoadsorption on (1) the dynamics of SOFA score and (2) hospital length of stay
Please also mention which tool/software were used for the Forest plot interpretion, mention with parameters in the figure legeds.
Although the study has a high risk of bias, and failed to offer a high-grade recommendation for immunomod-Ulation, current evidence, however it may help in future studies to draw a definitive conclusion of using CytoSorb in critically ill septic patients.
Author Response
Dear reviewer,
Thank you very much for your input on our article, which has significantly improved our article. We hereby attach a point-by-point response to your comments.
Q1: Line 25. Please revise as “We did not observe any significant difference…”
R1: we have revised the sentence by adding “significant” to make the results more clear.
Q2: Line 62. Please edit the term Secondary” to “secondary”
R2: we have corrected the misspeling by replacing the upper case with a lower case.
Q3: Line 62. It would be much better if the timeline was set from 2014 to September 2024, or give a possible justification for time 2016-2024 selection.
R3: Our choice for the timeline has been explained in the methods section. Our meta-analysis was aimed at current definitions of sepsis and septic shock that were adhered to by SEPSIS-3 in 2016. This has been stated in lines 59-60. Also, the timeline was set to this timeframe to comply with current definitions (lines 77-78). We considered that patients treated before that date (from 2014 as you suggested) could not be that “critically ill” as previous to that time sepsis was only SIRS + infection. So haemoadsorption would have had significant benefits due to the SIRS component but with no organ dysfunction (if the patients did not have severe sepsis), they would not reflect the critical cases we now know as sepsis.
Q4: Please providet the primary outcome datasheet of meta-analysis for assesing the effects of hemoadsorption on mortality in septic patients, the secondary outcomes were to determine the effects of hemoadsorption on (1) the dynamics of SOFA score and (2) hospital length of stay
R4: The primary outcome can be found in the last column of table 1 (mortality outcomes). If the reviewer thinks something else would be more appropriate, please provide us with supplemental guidance.
Q4: Please also mention which tool/software were used for the Forest plot interpretion, mention with parameters in the figure legeds.
R4: The tool for statistical analysis has been mentioned in the methods (statistical analysis subchapter, lines 102-103) – “The statistical analysis was performed using the Rev-Man online tool (Cochrane Collaboration, Oxford, UK) available at https://revman.cochrane.org/.”. Also, in the same statistical section, readers can find the interpretation for the Forest plots. We consider that adding them to the figures would only make the article harder to read by repeating the same information in two places. Also, both the statistics and their interpretation is standard for a meta-analysis and no supplemental tests were used.
This manuscript is a resubmission of an earlier submission. The following is a list of the peer review reports and author responses from that submission.
Round 1
Reviewer 1 Report
Comments and Suggestions for Authors
Comments
I reviewed the article entitled "To hemoadsorb or not to hemoadsorb – do we have the answer yet? An updated meta-analysis on the use of CytoSorb in sepsis and septic shock" and found some major flaws and high duplication rate, particularly to the authors previous articles.
iThenticate report
According to the iThenticate report, which Cora Feng, uploaded on 2024-11-12 03:50:47, the article had 38% wording duplication. My searching show that the author has copied pasted their own previously published article. I believe that this percentage is too high and should be reduced to less than 15%. Some texts with a high duplication rate are shown below. Before resubmitting an article, the author should check the duplication rate. For detail, please see the pdf file attached to the "Comments and Suggestions for authors"

Comments on the Quality of English Language
copied/pasted data
Reviewer 2 Report
Comments and Suggestions for Authors
This is an interesting study and the reviewer noticed need to some minor correction.
1. According to the results, the authors chose ten studies, please describe the areas of the studies such as Europe, Asia , etc. This is also important issue about the effect of drug depends on the different populations.
2. Did the authors consider facility of care for the patients. it would be the confounding factor for the mortality rate, length of stay in the hospital. How did the authors reduce the confounding factor at this study?
3. Please also make brief discussion about the facility of the treatment center among the enrolled study
Reviewer 3 Report
Comments and Suggestions for Authors
The manuscript entitled "To Hemoadsorb or Not to Hemoadsorb - Do We Have the Answer Yet? An Updated Meta-Analysis on the Use of CytoSorb in Sepsis and Septic Shock" provides insight into the effects of CytoSorb on sepsis and septic shock. However, the manuscript's quality can be further improved by addressing the following queries:
1. Figures 1, 2, and 7 require improvement to meet the expected quality standards.
2. The caption title of Table 1 is incomplete and should be revised to accurately reflect the comparative studies presented.
3. Table 1: Kindly clarify the meaning of the asterisk mark (*) for improved comprehension.
4. In Figure 3, the data should be presented in a sequential order, either from the earliest year to the latest or vice versa, to enhance the clarity of the presentation.
5. Figure 3: Since ten studies are included in the analysis to examine trends using a forest plot, the corresponding details can be written or included in the caption title for clarity.
6. As the authors suggestive that the current research is not yet comprehensive or definitive, the authors should provide guidelines to achieve a conclusive outcome from this study.